# Comparative Study of the Effects of Light Controlled Germination Conditions on Saponarin Content in Barley Sprouts and Lipid Accumulation Suppression in HepG2 Hepatocyte and 3T3-L1 Adipocyte Cells Using Barley Sprout Extracts

**DOI:** 10.3390/molecules25225349

**Published:** 2020-11-16

**Authors:** Jae Sil Kim, Eunseon Jeong, So Min Jo, Joonho Park, Ji Yeon Kim

**Affiliations:** 1Department of Food Science and Technology, Seoul National University of Science and Technology, 232, Gongneung-ro, Nowon-gu, Seoul 01811, Korea; jsilk88@naver.com (J.S.K.); todayjung@hanmail.net (E.J.); mindar71@naver.com (S.M.J.); 2Department of Fine Chemistry, Seoul National University of Science and Technology, 232, Gongneung-ro, Nowon-gu, Seoul 01811, Korea; jhpark21@seoultech.ac.kr; 3Department of Nano Bio Engineering, Seoul National University of Science and Technology, 232, Gongneung-ro, Nowon-gu, Seoul 01811, Korea

**Keywords:** barley sprout, flavonoids, germination conditions, lipid accumulation, polyphenol

## Abstract

Barley sprouts (BS) contain physiologically active substances and promote various positive physiological functions in the human body. The levels of the physiologically active substances in plants depend on their growth conditions. In this study, BS were germinated using differently colored LED lights and different nutrient supplements. Overall, there were 238 varied BS samples analyzed for their total polyphenol and flavonoid contents. Principal component analysis (PCA) was performed to determine the relationship between the germinated samples and their total polyphenol and flavonoid contents, and those with high levels were further analyzed for their saponarin content. Based on the PCA plot, the optimal conditions for metabolite production were blue light with 0.1% boric acid supplementation. In vitro experiments using the ethanol extract from the BS cultured in blue light showed that the extract significantly inhibited the total lipid accumulation in 3T3-L1 adipocytes and the lipid droplets in HepG2 hepatocytes. These findings suggest that specific and controlled light source and nutrient conditions for BS growth could increase the production of secondary metabolites associated with inhibited fat accumulation in adipocytes and hepatocytes.

## 1. Introduction

Sprout vegetables have been eaten in China for over 5000 years. In Korea, bean sprouts and radish sprouts are representative of vegetable sprouts, and in Europe and Australia, sprout vegetables are consumed by more than 30% of the population [1]. Sprout vegetables are generally defined as young vegetables that were sprouted from seeds for approximately one week and have 1–3 leaves. They are less than 10 cm in size, which is smaller than full grown vegetables, but contain 5–10 times more nutrients [2]. With optimum germination conditions such as light, nutrients, and moisture, the plant sprouts produce bioactive material to defend themselves from external attack [3].

Additionally, these bioactive materials are beneficial for human consumption [2,4,5]. When the leaves are young, the amount of these useful physiologically active substances is maximum in the cotyledons, and the young leaves contain 4–100 times more of these substances than those in fully grown plants [1,6].

Barley (*Hordeum vulgare* L.), belonging to the family Poaceae (Gramineae), is an annual plant, and it is currently one of the four main global cereal crops [5].

It is one of the oldest food crops and the second-most consumed grain, after rice, in Korea [2].

According to a recent report, barley sprouts (BS) promote various positive physiological functions in the human body, increasing the interest in their research [7,8]. In addition to typical flavonoids and polyphenols such as saponarin (SA), policosanols, and lutonarin, BS also contains physiologically active substances such as quercetin, kaempferol, catechin, and β-glucan (a water-soluble dietary fiber known to have cholesterol-lowering properties) [5]. Previous studies reported preventative effects associated with these compounds in metabolic diseases like hypertension, obesity, and diabetes [2,9].

BS also contains physiologically active substances that are necessary for bud growth, which shown increased production when cultivated under different germination conditions [10,11].

As for research to increase physiologically active substance, research using LED light water condition, nutrients, and research on germination conditions are being conducted [4,11,12].

In particular, saponarin was the sole flavonoid in the primary leaves of barley sprout [11]. Under the blue light treatment, the contents of saponarin and the genes related to the biosynthetic pathways, chalcone synthase, chalcone isomerase, UDP-Glc: isovitexin 7-O-glucosyltrasnsferase, were highly induced during the development of young leaves [11,13]. These results suggested that de novo synthesis of saponarin was increased under blue light [11,12,14]. Therefore, it was expected that BS grown in blue light conditions would increase flavonoids such as saponarin.

SA (apigenin-6-C-glucosyl-7-*O*-glucoside), also known as a flavanone glucoside, is the main flavonoid in barley and is found in several other plants [15].

SA was found to influence the maintenance and regulation of AMP-activated protein kinase (AMPK)—a protein that promotes phosphorylation. It increased intracellular calcium levels and induced AMPK phosphorylation, thereby increasing endurance and improving cholesterol via inhibition of fat and cholesterol biosynthesis processes [16,17]. Based on these results, it could be seen that saponarin activated AMPK in a calcium-dependent manner [16]. Another study showed that BS could treat hyperlipidemia by inhibiting the activity of the HMG-CoA (3-hydroxy-3-methyl-glutaryl-CoA) enzyme, which is involved in hexacosanol cholesterol biosynthesis [9].

Studies on optimal growth conditions to increase the production of secondary metabolites such as saponarin in BS are limited. In this study, conditions that might increase the production of secondary metabolites during BS germination were investigated, and the inhibitory effects of saponarin on 3T3-L1 cells were evaluated using an ethanol extract of BS cultivated under optimal conditions. The BS ethanol extract was also evaluated for its potential to mitigate fatty lipid accumulation in HepG2 cells.

## 2. Results

### 2.1. Total Flavonoid and Polyphenol Content and PCA Analysis

The total flavonoid and polyphenol contents were quantified from 238 samples to investigate the effects of their growing conditions on secondary metabolite production. These samples were grown under three different LED light sources (blue, white, or red) with different water types (hydrogen water, tab water), supplements (Bo, Zn, Se, Si), and range of concentration (0.1–0.00001%), at different growth temperatures (4 °C, 25 °C, 40 °C, 60 °C) and for different temperature exposure times (10 min, 60 min) at germination. Hydrogen water was set as a control group.

The total flavonoid and polyphenol contents of the 70 samples grown in the red light were 0.80–14.54 and 0.51–7.68 mg/g, respectively; whereas for the 78 samples grown in white light, these were 1.03–16.06 and 1.18–124.62 mg/g, respectively; for the 70 samples grown in blue light, these were 2.82–42.72 and 1.68–75.38 mg/g, respectively, and for the 20 samples grown in white and blue light, these were 25.76–31.76 and 5.2–20.75 mg/g, respectively (Figure 1A).

Principal component analysis (PCA) was performed to determine the relationship between the germinated samples and the secondary metabolite content. To observe the changes in the total polyphenol, flavonoid, and SA contents, in response to the germinated conditions such as light, water, supplement, stress temperature, and time conditions, an SIMCA-P+ analysis of the PCA was conducted. It revealed an association between the light conditions and secondary metabolite production. Based on the secondary metabolite results, the PCA showed that the x–y axis was generated, where the *x*-axis was the first principal component, and the *y*-axis was the second principal component. The first 238 samples were cultivated under four different light conditions, and the PCA was derived for the total polyphenol and flavonoid data. The results showed a 27.5% positive influence for white light, which was the second principal component, and a 35.0% positive influence for the blue and white + blue light, which was the first principal component (Figure 1B)

### 2.2. Saponarin Content and PCA Analysis

The samples selected received specific nutrient supplementation with boric acid or zinc, under white or blue light, and yielded high levels of secondary metabolites (Figure 1C). Excluding the samples grown in the red light, 72 samples that had higher levels of secondary metabolites were subjected to additional SA assays and PCA for polyphenol, flavonoid, SA, and light conditions. As a result, a 46.6% influence in the positive direction for blue light (the first principal component), and a 39.1% influence in the positive direction for white + blue light (the second principal component) was observed. It was also found that the positive influence of the blue light had the same effects as the SA and the total flavonoid contents (Figure 1D). Among these sprout samples, the 48th sample had the highest SA content; it was grown in blue light with 0.1% boric acid supplementation. Furthermore, the total polyphenol, flavonoid, and SA contents for this sample were 53.21 mg/g, 35.50 mg/g, and 11.14 mg/g, respectively.

Figure 2 shows the profiles of standard compounds and the identification of SA in barley sprout extracts (BSE). The HPLC analysis of SA revealed a peak, matching the standard commercial retention time of approximately 10.3 min (Figure 2). The SA content was 1.66–11.14 mg/g. The results of the samples with top five SA content from the BS was 7.88–11.14 mg/g. The sample with the highest saponarin content was sample 48, grown under the conditions of blue light and 0.1% boric acid. Their conditions were as follows—two samples were supplied with boric acid 0.1%, and the others were each supplied with selenium 0.1%, boric acid 0.001%, and hydrogen water; all samples were under blue light. The results of the lower five samples for SA content in the BS were 1.81–2.09 mg/g. These conditions were as follows—three samples were cultivated in hydrogen water, two samples were grown in tap water, and all samples were under white light.

### 2.3. Effect of BSE on Fat Contents in 3T3-L1 and HepG2 Cell Lines

Before treating 3T3-L1 and HepG2 cells with BSEs, we first analyzed the cytotoxicity of the experimental concentrations on these two cell lines. The 3T3-L1 and HepG2 cells were exposed to different concentrations of SA (8, 25, 50, and 100 μM) or BSE (500 μg/mL). In the 3T3-L1 and HepG2 cells, the treated concentrations of SA did not affect cell viability, nor did the 500 μg/mL BSE; these concentrations were thus used in subsequent experiments (data not shown).

We examined the effects of BSE and SA on adipogenesis, by adding samples to the medium every 3 days during cell differentiation. To measure the adipogenesis, the cells were stained with oil red O on day 12 of the differentiation and then microscopically observed. After microscopic observation, the cells were dissolved in 2-propanol and compared with the control group. The absorbance values of the lipid droplets were determined as relative values. The average absorbance value in the control group was 0.25 ± 0.06, whereas that for the lipid droplets cultured for 12 days in the differentiation medium was 0.67 ± 0.02, indicating that fat accumulation increased by 267.02% and that differentiation progressed sufficiently. In the sample group, lipid droplets were significantly decreased in all samples, except at 8 μM SA, which was the concentration of BSE. At 50 μM SA, lipid droplets decreased by 45.97%, and in 100 μM SA, lipid droplets decreased by 57.32%. In BSE, the decrease was by 20.12% (Figure 3A,B).

HepG2 cells were treated with oleic acid to induce NAFLD, and oil red O staining was used to measure the potential inhibitory effects of the BSE and SA on fat accumulation. In the oleic-acid-treated group, fat accumulation was significantly increased, compared with the control, indicating that oleic acid induced significant fat accumulation. In the groups treated with BSE or SA, fat accumulation was significantly lower than that in the oleic acid treatment-only group, suggesting that lipid accumulation reduction was caused by the addition of BSE or SA. In all groups, lipid accumulation was evaluated by oil Red O staining. The average absorbance value for the control group was 0.16 ± 0.01, whereas that for the oleic-acid-only group was 0.28 ± 0.01, indicating a fat accumulation increase of 170.59%. In the BSE and SA sample groups, lipids were significantly decreased, except for 8 μM SA. At 25 μM SA, there was a lipid decrease of 4.99%; at 50 μM SA, there was a decrease of 14.85%; and in BSE, there was a decrease of 14.74% (Figure 3C,D).

## 3. Discussion

Sprout vegetables contain a diverse array of phytochemicals to protect themselves from external attack; levels of these physiologically active substances could change in response to the light energy received during germination [3]. BSE was reported to have many health-promoting properties, including antioxidant [16,18,19], anti-inflammatory [3], anti-obesity effects [20], as well as positively influencing cholesterol and blood flow [2]. Polyphenols and flavonoids contain many hydroxyl groups (-OH), which might exert their antioxidant effects by binding to (scavenging) reactive oxygen species (ROS) [21].

The levels of physiologically active substances in plants depend on their growth environment. Previous studies on cultivation environments in which specific wavelengths and nutrients of plant systems were formulated are insignificant [22].

This study examined different conditions for enhancing the antioxidant capacity of BSE, which germinated under different light and nutrient treatments. Germination conditions, including the temperature and nutrient supplementations, which might increase the SA content and the main active substance in BS, were also investigated. This study also found that BSE was associated with reduced fat accumulation in adipocytes and hepatocytes.

Extracts from sprouts grown under blue, red, or white light were investigated for flavonoid and polyphenol contents. Samples with high levels of these components were further analyzed for SA content; then, in vitro experiments on 3T3-L1 and HepG2 cell lines were carried out using BSE from plants grown under the conditions that yielded the highest SA content.

Comparing the levels of total flavonoid and polyphenol from barley germinated under the blue, white, or red light sources, demonstrated that the total polyphenol content was higher in blue light samples and lower in the red light samples.

Therefore, in this study, as a result of analyzing TP and TF of samples grown under 238 conditions, it was confirmed that TP and TF were increased with SA in the samples grown under blue light, among germination conditions. SA was selected as a representative substance among flavonoids in sprout barley, and 72 conditions that were expected to increase SA were selected for further analysis.

Additional experiments were conducted by adding the essential plant growth nutrients boric acid or zinc [23,24,25]. The SA content was 5.80 mg/g and 7.22 mg/g, in the white light samples treated with 0.1% zinc or boric acid, respectively. The SA content of the blue light samples treated with 0.1% zinc or boric acid was 7.65 mg/g and 11.14 mg/g, respectively. These results demonstrated that boric acid increased secondary metabolite production more than zinc [26], but PCAs of total flavonoids, polyphenols, and SA content, according to the different light, nutrient, or temperature conditions indicated that secondary metabolites were most strongly affected by the light conditions [27].

As such, in vitro experiments were carried out with extracts prepared using the BS grown under blue light to promote high SA content. The in vitro adipocyte and hepatocyte models were established using 3T3-L1 and HepG2 cells [28,29,30]. Previous studies showed that 3T3-L1 cells differentiate into adipocytes when induced with lipid differentiation media consisting of insulin, dexamethasone, and isobutyl methylxanthine. Differentiation into adipocytes leads to regulation of genes that are expressed specifically in adipose tissue cells; these factors are expressed in the differentiation process [31]. In the case of NAFLD, HepG2 cells were treated with FFAs, to observe the changes in their accumulation of triglyceride (TG) [32].

TG is a complex of FFAs and glycerol, and more than 95% of TG consumed as food is stored as fat, whereas the remaining TG is used as energy; over-consumption of foods containing TG can thus lead to obesity. Obesity is known to increase the risk of metabolic diseases such as hypertension and diabetes, and NAFLD was also reported to be closely related to obesity. Fat accumulation in the liver occurs when TG levels in the liver increase or TG emissions decrease; most TGs are supplied by circulating FFAs in the blood [33,34,35]. Concentrations of intracellular triglyceride in 3T3-L1 adipocytes and HepG2 hepatic cells were observed by staining with oil red O, to observe the inhibitory effects of the experimental concentrations of SA and BSE. Lipid accumulation was significantly inhibited at all SA concentrations except at 8 μM, which was the concentration of BSE.

Previous studies showed that AMPK plays an important role in energy metabolism. It inhibits the expression of SREBP-1c and PPARγ, which are associated with lipid metabolism, the expression of lipogenic enzymes such as FAS, and triglyceride production, and promotes fatty acid β-oxidation. Increased expression of AMPK increases the fatty acid oxidation in the liver, reduces fatty acid biosynthesis, increases fat degradation in adipose tissues, and decreases lipid synthesis. Thus, regulating AMPK and inhibiting the accumulation of triglycerides are an important factor in controlling metabolic diseases [36,37].

Particularly in some studies, saponarin is the major flavonoid (1.1%, *v/v*) in barley, and regulates glucose uptake by stimulating AMPK, an important regulator of energy balance [17]. In addition, the long-term intake of high-fat foods induces insulin resistance, as blood glucose levels are persistently elevated. BS intake might improve metabolic syndromes associated with obesity, by decreasing insulin concentration.

Therefore, based on previous investigations and the results of the present study, BS appear to improve AMPK through the regulation of its germination condition and increasing its secondary metabolite and saponarin content to suppress triglyceride accumulation. The present study confirmed that BS and saponarin could potentially ameliorate TG accumulation [20].

Additionally, BSE was also found to be effective in protecting nerve cells and improving blood circulation. The ethanol extract of the BS was reported to reduce superoxide dismutase (SOD) activity and levels of MDA, a lipid peroxidation product, due to oxidative stress in the neurons. In addition, it was confirmed that a hot-water extract of BS significantly delayed the activated partial thromboplastin times (aPTT) associated with platelet aggregation-related serotonin and anticoagulant activity [3,7,8]. These results were proposed to be due to flavonoids and polyphenols such as SA in the BS, as these compounds were previously shown to have antioxidant, anti-obesity, and anti-interstitial lipidation effects [18,38]. This research was conducted to identify the conditions that would increase the content of secondary metabolites such as flavonoids and polyphenols in BS. The antioxidant content varied across different parts of the barley plant and were the highest in barley leaves; several studies on barley leaves were carried out. In other studies, total polyphenol and total flavonoid contents of the BSE were reported to be 17.55 ± 0.53 and 13.98 ± 6.91 μg/mg, respectively, and the DPPH radical and nitric oxide scavenging activities were also reported.

In this study, various germination conditions examined to the increase in saponarin contents in barley sprouts, to study the effect of inhibiting fat accumulation using 3T3-L1 and HepG2 cells.

To determine the optimized environmental factors, such as light, water, nutrients, and temperature in germination and heat stress before harvesting, a total of 238 conditions were tested. PCA analysis was performed after analyzing total flavonoids and total polyphenols as secondary metabolites of 238 samples grown by light, water, nutrients, and temperature in germination and heat stress.

A result of PCA analysis in secondary metabolites indicated that quality of light affected the significant changes of those concentrations. Therefore, for further analysis of changes of saponarin concentration, three light conditions used were blue, blue + white, and white light. However, the test under the red light were excluded due to a tendency of low secondary metabolite content.

As a result of PCA analysis, it was confirmed that this was closely related to saponarin and blue light. Of the 72 saponarin results, the top 5 saponarin results were found to be 7.88–11.14 mg/g. Of these 5 samples, 3 were samples in which boric acid was added by concentration (two were 0.1% boric acid, and the other was 0.001%) in blue light, the other was in white light with Selenium 0.1%, and one was in blue light with hydrogen water. Saponarin results of the lower five samples were measured to be 1.66–11.14 mg/g, and all samples were grown with hydrogen water or tap water in white light (three were hydrogen water and the other two were tap water).

This study revealed that light color and nutrients during barley sprout germination could affect SA production and its resultant antioxidant effects on cells. In vitro experiments demonstrated that SA, an important component of BS, cultivated in blue light with boric acid, might be promising as an effective therapeutic drug for obesity mitigation, and that it is necessary to clarify efficient cultivation conditions to increase this efficacy.

## 4. Materials and Methods

### 4.1. Materials

Dulbecco’s modified Eagle’s medium (DMEM), fetal bovine serum (FBS), and penicillin–streptomycin were purchased from Biowest (Nuaille, Cholet, France). Bovine serum was obtained from Gibco (Rockville, MD, USA). Insulin, dexamethasone, 3-isobutyl-1-methylxanthine (IBMX), and MTT were obtained from Sigma-Aldrich (St. Louis, MO, USA).

### 4.2. Cultivation of Barley Sprout

Seed of Korean barley (*Hordeum vulgare*) were purchased from Danong. Co., Ltd. (Seoul, Korea). BS was harvested after 9 days of germination (i.e., on the 10th day). Sprouts were cultivated under the conditions described in Table 1, to examine the effect of the type and concentration of the nutrients on the saponarin and total polyphenol content. Hydrogen water was set as a control group, and the BS were treated with 0.01–0.000001% concentration of Zn, Bo, Si, and Se, and then each was exposed to blue, red, and white light. It was classified into 70 blue lights, 20 blue + white lights, 78 white lights, and 70 red lights. A total of 238 varied samples were harvested. All samples were freeze-dried and stored at −80 °C until analysis.

### 4.3. Total Polyphenol and Flavonoid Content

The total polyphenol contents were evaluated using the Folin–Ciocalteu assay [39]. Either the sample or the gallic acid standard was mixed with Folin–Ciocalteu and diluent water for 5 min. Subsequently, 10% sodium carbonate was added to the reaction mixture, and the solution was incubated for 30 min at room temperature. The absorbance was measured at 750 nm using a spectrophotometer (BioTekInstruments, Inc., Winooski, VT, USA). The results were expressed as mg of gallic acid equivalent/g sample. The total flavonoid content in each BSE was measured using the aluminum chloride colorimetric method with catechin [40]. The sample or catechin was mixed with 5% sodium nitrite and diluent water. After 5 min at room temperature, 10% aluminum chloride was added for 6 min. Next, 1 M sodium hydroxide was added with diluent water, and the absorbance of the solution at 510 nm was immediately measured. The total flavonoid content was expressed as mg/g of the sample.

### 4.4. Quantification of Saponarin Using High-Performance Liquid Chromatography (HPLC)

HPLC analyses were conducted with a Shiseido SI-2 series HPLC system (Shiseido, Tokyo, Japan), equipped with a Shiseido Capcell Pak C18 MG II S-5 (250 × 4.6 mm; i.d., 5 µm) column. The mobile phase consisted of water (solvent A) and acetonitrile (solvent B), both containing 1% formic acid, and was applied as follows: 0–20 min, 0–10% (B), 21–31 min, 10–40% (B), and 32–35 min, 40–60% (B), followed by re-equilibration with 10% (B) for 36–42 min, at a flow rate of 1.0 mL/min and an injection volume of 5 µL. The detection wavelength of SA was set to 254 nm. The SA content in the BSE was verified by comparing the retention times to the external standards. Stock mixed standard solutions with SA were prepared in methanol at the following concentrations: 0, 6.25, 12.5, 25, 50, and 100 µg/mL. The BSEs were immersed in 300 µL methanol, filtered, and injected into the HPLC system.

### 4.5. Preparation of Barley Sprout Extracts

BS to be analyzed for their total flavonoid and total polyphenol were prepared by adding a certain amount of the sample to ethanol and then extracted for 24 h. The extract was then used for analysis after centrifugation. BSE was prepared by adding a certain amount of sample to 70% ethanol at 1:20 (*w/v*) and stirring for 24 h. The extract was filtered with filter paper, concentrated at 55 °C in a rotary evaporator (UT-1000; EYELA, Tokyo, Japan), and then freeze-dried for use as an ethanol extract.

### 4.6. Cell Culture

3T3-L1 cells (mouse fibroblast 3T3-L1 preadipocyte) and HepG2 cells (human liver hepatocellular carcinoma cell line) were obtained from the Korean Cell Line Bank (Seoul, Korea). The HepG2 cells were used to confirm the inhibitory effects on lipid accumulation in the liver, and the 3T3-L1 cells were used to examine the effects of the saponarin on adipocyte differentiation. The 3T3-L1 cells were cultured in DMEM supplemented with 1% penicillin and 10% BS. The HepG2 cells were cultured in DMEM supplemented with 2% penicillin, 2% HEPES, and 10% FBS. Cell cultures were incubated in a 5% CO2 humidified incubator at 37 °C. The medium was changed every 2–3 days, and the cells were subcultured at approximately 80–90% confluency. To determine the effects of the samples on cell viability, a cell viability assay was performed using an MTT assay. The cells were seeded in a 96-well plate at a concentration of 1 × 10^5^ cells/mL and cultured at 37 °C in humidified air and 5% CO_2_ for 24 h. The cells were treated with various concentrations of BSE for 24 h. The cells were then added to the MTT solution and incubated for 2 h to produce formazan. The formazan crystals were dissolved in dimethyl sulfoxide, and their absorbance was measured at 560 nm using a microplate reader (Biotek Winooski, VT, USA). The applied concentrations of the extracts were not cytotoxic to the 3T3-L1 and HepG2 cells.

The 3T3-L1 pre-adipocytes were seeded at 1 × 10^5^ cells/mL in 24-well plates with the culture medium. The cells were plated at a density that allowed them to reach confluence in 3 days. The day when cell differentiation was initiated was designated as Day 0. At this point (Day 0), the cells were switched to MDI-differentiation medium (DMEM, 10% FBS, 10 μg/mL insulin, 115 μg/mL IBMX, and 1 μM dexamethasone) for 6 days. The medium was changed to DMEM containing 10 μg/mL insulin 6 days later. The medium was changed every 3 days. To elucidate the effects of saponarin on adipogenic differentiation, saponarin was added to the 3T3-L1 cells on Day 0. The same concentration of saponarin was added at 3-day intervals when the culture medium was replaced. After 12 days, the cells were harvested.

HepG2 cells were seeded at 1 × 10^5^ cells/mL in 24-well plates with the culture medium. When the cell density reached approximately 90%, the medium was replaced with serum-free DMEM, and for the induction of lipoapoptosis, samples were treated with free fatty acids for 24 h. Oleate stocks (100 mM) were prepared in distilled water at 70 °C. A 5% (*w/v*) free fatty acid (FFA)-free BSA solution was prepared in serum-free DMEM, and then a 2.5 mM FFA/5% BSA solution was prepared by complexing an appropriate amount of FFA to 5% BSA in a 40 °C water bath.

### 4.7. Oil Red O Staining

The fat accumulation in the cells was qualitatively and quantitatively analyzed by oil red O staining. The cells were washed with DPBS and fixed in 4% paraformaldehyde for 60 min at 4 °C. The fixed cells were then stained with 0.5% oil red O in a working solution that was diluted with isopropanol (6:4) in distilled water for 30 min at room temperature, without light, and then washed twice with distilled water. Lipid droplets were stained and observed using light microscopy (Nikon, Tokyo, Japan). After observations with a microscope, the dyed lipids in the cells were dissolved in 1 mL of 2-propanol for 10 min, and the absorbance was measured at 560 nm, using a microplate reader (Biotek Winooski, VT, USA).

### 4.8. Statistical Analysis

Data are expressed as the mean ± standard deviation (SD). Statistical analyses were performed using Duncan’s multiple range test of one-way analysis of variance (ANOVA) (SPSS 20; SPSS Inc., Chicago, IL, USA). *p* < 0.05 was considered to be statistically significant. PCA-X data were analyzed using partial least square discriminant analysis (SIMCA-P +, PLS-DA).

## Figures and Tables

**Figure 1 molecules-25-05349-f001:**
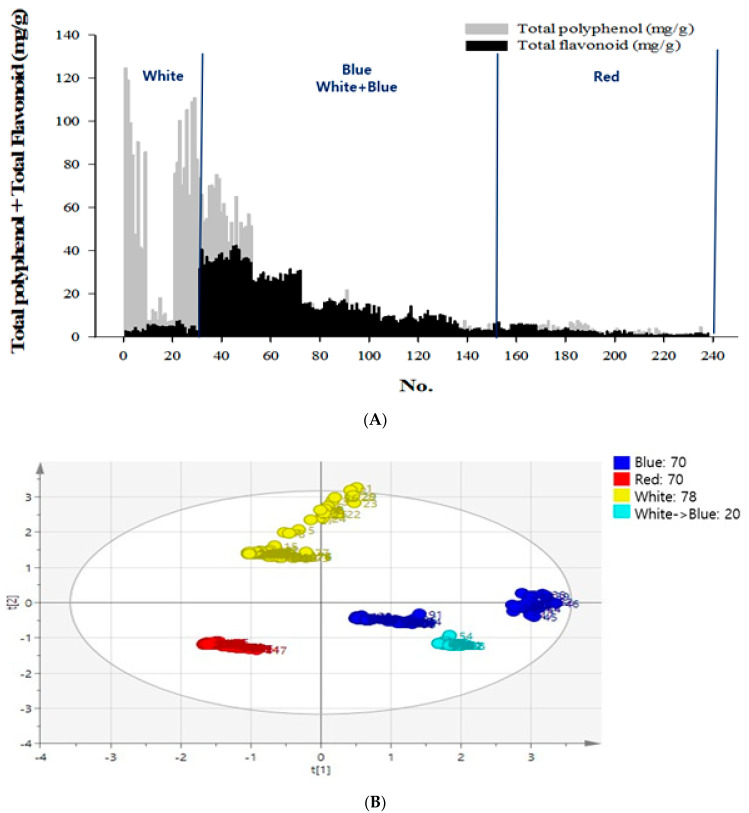
Contents of chemical compounds—flavonoid, polyphenol, and saponarin and principal component analysis (PCA) of the secondary metabolites in barley sprouts (BS). (**A**) Total flavonoid and polyphenol contents of the 238 samples cultivated in red, white, or blue light. (**B**) Total polyphenol and flavonoid contents of the 238 samples grown under different LED light sources. (**C**) The conditions of 72 samples selected were analyzed for total flavonoid, total polyphenol, and saponarin. (**D**) The total polyphenol, flavonoid, and saponarin contents for the 72 samples were selected from the nutrient supplementation groups with boric acid and zinc under white light or blue light. Test sample was one and data were expressed as the mean of three independent experiment, and then PCA analysis was performed with the mean value.

**Figure 2 molecules-25-05349-f002:**
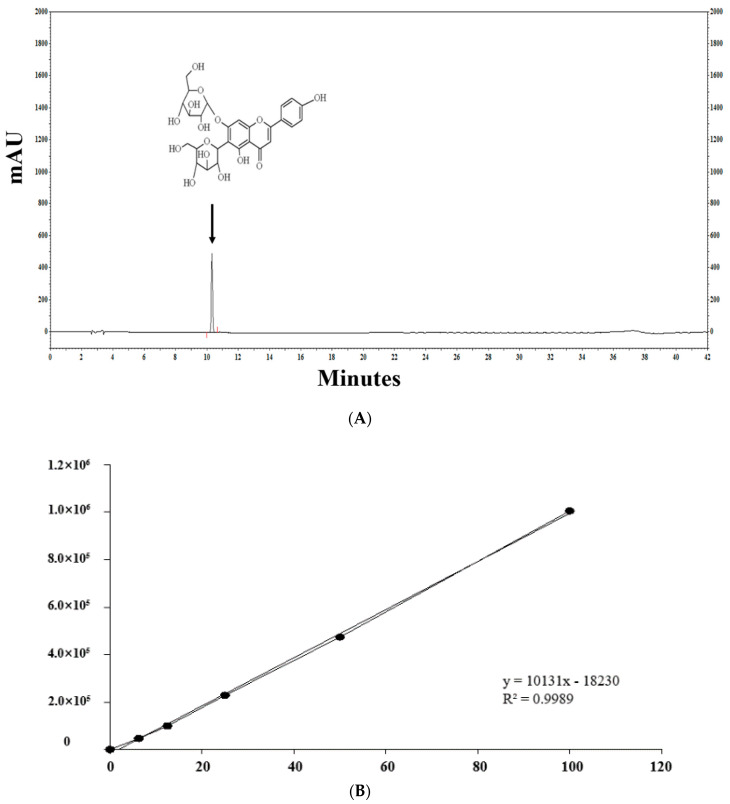
HPLC chromatogram of the standard and barley sprout extracts (BSE) and a graph of the top and bottom five BS samples for saponarin content. HPLC chromatogram standard (**A**) and standard quantitative curve (**B**), and HPLC chromatogram of the BS samples with the highest (**C**) and the lowest saponarin contents (**D**). Saponarin appeared with retention times of approximately 10.3 min at 254 nm.

**Figure 3 molecules-25-05349-f003:**
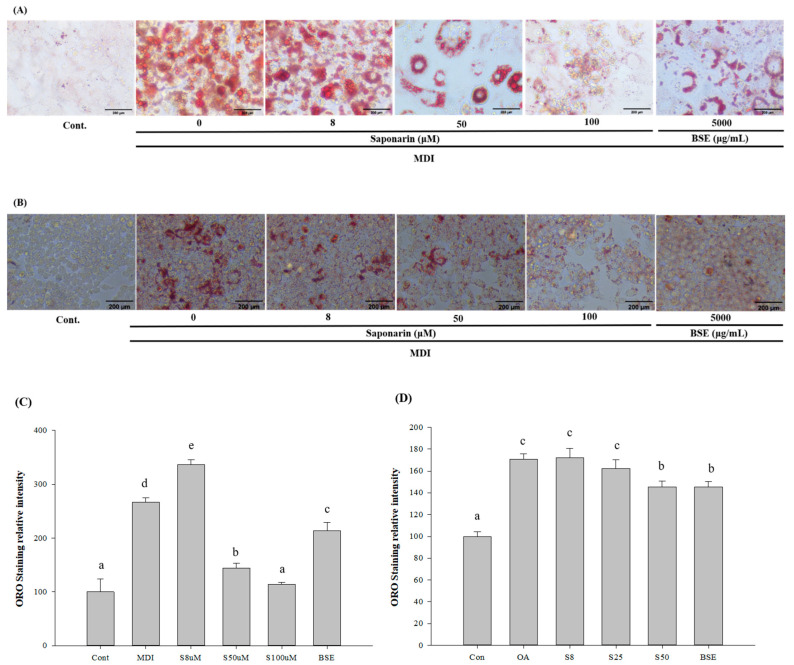
Effect of barley sprout extracts (BSEs) and saponarin on fat accumulation in the 3T3-L1 cells and HepG2 cells. (**A**) Oil red O staining of the intracellular triglycerides in the 3T3-L1 cells. The 3T3-L1 cells were treated with BSE (500 μg/mL) or saponarin (0, 8, 50, and 100 μM) during differentiation. The assays were performed on fully differentiated adipocytes (day 12). The selected images of three independent experiments. (**B**) Oil Red O staining of the intracellular triglycerides in the HepG2 cells. HepG2 cells were treated with BSE (500 μg/mL) or saponarin (0, 8, 25, and 50 μM) during fat accumulation (24 h). The selected images of three independent experiments. (**C**) Relative density of the lipid contents in the 3T3-L1 cells. (**D**) Relative density of the lipid contents in the HepG2 cells. Statistical analyses were performed using Duncan’s multiple range test one-way analysis of variance (ANOVA) (a, b, c, d, e). Data are expressed as mean ± SEM of three independent experiments. abc: Different superscripts indicated mean difference between groups significant at *p* < 0.05.

**Table 1 molecules-25-05349-t001:** Conditions of the barley sprout (BS) germination treatments.

Light	Water	Temperature	Hour	Nutrition	Concentration
Blue	Hydrogen Water	Control	-	-	-
		4 °C	10 min	-	-
			60 min		
		25 °C	10 min		
			60 min		
		40 °C	10 min		
			60 min		
		60 °C	10 min		
			60 min		
		-	-	Zn	0.1
					0.01
					0.001
					0.0001
					0.00001
				Bo	0.1
					0.01
					0.001
					0.0001
					0.00001
	Tab water	4 °C	10 min	-	-
			60 min		
		25 °C	10 min		
			60 min		
		40 °C	10 min		
			60 min		
		60 °C	10 min		
			60 min		
Blue + White	Hydrogen Water	Control	-	-	-
		-	-	Zn	0.001
				Bo	0.001
					0.0001
White	Hydrogen Water	Control	-	-	-
		-	-	Zn	0.1
					0.01
					0.001
					0.0001
					0.00001
				Bo	0.1
					0.01
					0.001
					0.0001
					0.00001
				Se	0.1
					0.01
					0.001
					0.0001
					0.00001
				Si	0.1
					0.01
					0.001
					0.0001
					0.00001
	Tab water	Control	-	-	-
		4 °C	10 min	-	-
			60 min		
		25 °C	10 min		
			60 min		
		40 °C	10 min		
			60 min		
		60 °C	10 min		
			60 min		
Red	Hydrogen Water	4 °C	10 min	-	-
			60 min		
		25 °C	10 min		
			60 min		
		40 °C	10 min		
			60 min		
		60 °C	10 min		
			60 min		
	Tab water	4 °C	10 min		
			60 min		
		25 °C	10 min		
			60 min		
		40 °C	10 min		
			60 min		
		60 °C	10 min		
			60 min		

(Triple test for each condition).

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
