# Peer review of "Comparative Study of the Effects of Light Controlled Germination Conditions on Saponarin Content in Barley Sprouts and Lipid Accumulation Suppression in HepG2 Hepatocyte and 3T3-L1 Adipocyte Cells Using Barley Sprout Extracts"

_molecules, 2020, doi:10.3390/molecules25225349_

Round 1

Reviewer 1 Report

The research article entitled “Comparative study of the effects of germination conditions on saponarin content in barley sprouts and lipid accumulation suppression in HepG2 hepatocyte and 3T3-L1 adipocyte cells using barley sprout extracts” is a good attempt to identify the growth conditions of barley sprouts in a comparative manner and the effects on lipid accumulation as an alternative supplement with properties that improve human health.  

However, major concerns have been raised over the content of the manuscript, and I have now recommended major revisions mainly about the lack of details of methods, treatments, and statistical analysis. It is hard to tell from the figures, results and methods the number of observations and treatments that went into each analysis, how many samples were analysed for each dataset, etc.

My questions and suggestions are as follows:

The title is too large and not very concise; I think that the title could be improved.

Page 1, line 17: “physicologically” is not properly written.

Lack of details in Introduction about the background respect to the effects of coloured LED lights, nutrient supplements, and temperature on production of secondary metabolites in sprouts. Please provide some evidence about it.

Example: Samuoliene et al. 2011. The impact of LED illumination on antioxidant properties of sprouted seeds

Results

Figure 1. Please detail the X-axis in A and C and identify the groups of samples that grown under different treatments (light conditions) in Fig. 1A and C. What are the conditions of Control group? Why the treatments did not have the same number of samples? Please, provide a description of two groups of samples grown under blue light (Fig 1B).  Please, improve the quality of graphs, the legends and axis names, these should be reading clearly.  Add in the figure legend the “n” of samples of each treatment.

Page 3, line 111: “BSE” is not mentioned before in the text, please add: “barley sprout extracts (BSE)”, and then use the abbreviation form.

Figure 2. The axis can not be read. Please, improve the quality of the figure. In Fig 2B which chromatogram correspond to the highest and the lowest saponarin contents? Please, identify it in the figure, and add the axis. What is the data that is being shown in the Fig 2C (is it “mean values” with the standard error or standard deviation, from what values, which is the “n” of reps etc). This needs to be clearly stated in the legend.

Figure 3. How was the statistical analysis done? Please, add in the legend what type of statistical analysis you performed and what is the meaning of groupings identified by letters. Add the information of time in which the cells were analysed (at 12 days of culture?), specify what was the conditions of control group cells. Are the control conditions in Fig. 3A the same that in Fig.3C? Why the control was identified as “undiff.” In Fig. 3A and “Cont.” in Fig 3C? How many images by condition were analysed?. You need to add the scale bar in the images that represent the magnification of images were acquired. Please, provide all the information in the figure legends and in the materials and methods section.

Methods

What cultivar of barley was used in this study? Did you use the same cultivar for all treatments?

Detail what a sample consists, maybe one sample is a group of 5, 10, or “n” sprouts? Or individual sprout? It should be clear in the materials and methods section and in legend of figures.

What means 238 varied samples?  how many samples were analysed for each dataset or treatments? This needs to be clearly stated so that one can understand how many values are going into each observation and also it needs to be explicitly stated if the values are subsamples and where analysed as such in the ANOVA or statistical analysis.

The treatments are not clearly shown in table 1. Please, add the control treatment conditions, and identify each treatment.

Discussion

Page 6, Lines 184-186: Please comment on the fact that blue light increases the SA synthesis. Support this with evidence.

For a better interpretation of figures and results, please describe well the treatments in table 1, dividing each treatment such as treatment 1, 2, 3 and identify this in each figure. It is hard to tell from the figures the number of observations and treatments that went into each analysis, how many samples were analysed for each dataset, etc.

Page 6, line 201. TG is the abbreviation of triglyceride? Please clarify all the abbreviations when it is used by a first time in the text.

Discuss more about the relationship between growth conditions (light, temperature, nutrients) and polyphenols, flavonoids and/or saponarin content. Please summarize the main findings of the study.

Author Response

Thank you for all of your comments. I have revised them, please check the attached file. Thank you.

Reviewer 2 Report

The manuscript deals with a very important and actual topic of the effect environmental conditions (mainly light quality and nutrients)  on the accumulation of total polyphenols and specifically saponarin in young barley sprouts. The study had a very complex design with a total of 5-6 (not clear how many?) environmental factors, but the experimental design and cultivation methodology are not sufficiently explained. It is constantly repeated that a total of 238 samples were obtained, but it is not described here whether the effect of light was monitored alone, under certain specific nutritional conditions and temperature, or in all combinations with other factors (then the number of samples should be higher). How many repetitions were for each variant/combination of factors (only 1 repetition?). It is not described what type of light source (probably LEDs) was used, what was the central wavelength, which PAR intensity was used (was it the same for all variants?), what was used as a white light source (LEDs consisting of RGB bands or a broadband source?). How did the cultivation actually take place? Hydroponically? This is absolutely essential information for the application of the results in practice and also for an eventual repetition of the experiment.
Why was only saponarin determined from all flavonoids? Although it is one of the most abundant, it is also the least affected by environmental conditions. On the contrary, lutonarine but also other flavonoids, including hydroxycinnamic acids, generally show a greater response to light or stress factors.
Similarly, the way the results are presented is very confusing and it is difficult to draw unambiguous conclusions from it. Although PCA shows a fundamental influence of the spectral composition of light on the content of flavonoids and saponarin, the direction and intensity of the influence is not apparent. The use of RDA could also include the influence of other environmental variables such as temperature or nutrition, which are not at all apparent from the graphs. It would also help if the graphs 1A and 1C indicated, for example, by a horizontal line the individual light treatments, nutrition, or other factors. The description of the effect of light and nutrition in result section is good, but these results cannot be verified at all from the graphs. What is the aim of graph 2C, for example, which shows only samples with the highest and lowest saponarin content?  It is much more important to show the average effect of individual treatments and, if possible, the interactions. The design of the experiment probably does not allow evaluation by multifactor ANOVA and post-hoc testing of differences between means. If so, it would be appropriate to show these results in a table, including interactions, and then present in the graph only the factor that most affected the content of polyphenols, flavonoids and saponarin.
In contrast, part of the testing of the effect of barley sprout extracts and saponarin on fat accumulation and triglycerides in the 3T3-L1 cells and HepG2 cells is presented in an unambiguous form, including the significance of differences. Only there is no description of the method by which the significance of the differences between the means was tested.
In the introduction, data without a literature source are given in several places… e.g. L. 38 contain 5-10 times more nutrients or L. 51… to have cholesterol-lowering properties.
A short introduction to other flavonoids and their role as antioxidants is also missing and the description devoted only to saponarin shows a one-sided view on the complex of flavonoids in barley.
The article is otherwise written quite clearly and comprehensibly, and the findings can also be very valuable. However, it is essential that the results are not clearly visible from the graphs and the graphs do not support the description in the text. Therefore, my evaluation is “major revisions”, while it is necessary to in details specify the methodology and change the way the results are presented in the graphs so that the conclusions written in the text are clearly supported.

Author Response

(The authors gave the same response as above.)

Round 2

Reviewer 1 Report

Comments and Suggestions for Authors:

The research article entitled “Comparative study of the effects of light-controlled germination conditions on saponarin content in barley sprouts and lipid accumulation suppression in HepG2 hepatocyte and 3T3-L1 adipocyte cells using barley sprout extracts” is a good attempt to identify the growth conditions of barley sprouts in a comparative manner and the effects on lipid accumulation as an alternative supplement with properties that improve human health. 

I Accept the manuscript after minor revision (corrections to minor text editing).

My suggestions are as follows:

I suggest the following title: “Comparative study of the effects of light-controlled germination conditions on saponarin content in barley sprouts extracts and their impact in lipid accumulation suppression in HepG2 hepatocyte and 3T3-L1 adipocyte cells”

Page 1, line 17: “physicologically” is not properly written. Please, change by “physiologically”.

Please, increase the font size of the axis names of Figure 2.

Author Response

Thank you for all of your valuable comments.

Reviewer 2 Report

The authors adressed most of my requirements. I understand that the identification of factors such as blue light in the Fig. 1  could be difficult. The table desribing the combinations of factors is very useful, although quite large. If necessary, it can be presented as a supplementary table. However, I am still missing some proof of the statements about the effect of blue light or other factors. If it is not possible to identify the numbers of samples presented in Fig 1  by the same numbers in the table of factors (it can be also used as a range eg. 10-15),  I would recommend at least to present bar graph as a part of Fig. 1 showing the means for key factors for phenolics, flavonoids and saponarin. This is my only requirement now, but crucial for clear presentation of results and better understanding to readers.   

Author Response

(The authors gave the same response as above.)
